# Food Waste and Its Association with Diet Quality of Foods Purchased in South Florida

**DOI:** 10.3390/nu13082535

**Published:** 2021-07-24

**Authors:** Vanessa Mijares, Jair Alcivar, Cristina Palacios

**Affiliations:** Dietetics and Nutrition Department, Robert Stempel College of Public Health & Social Work, Florida International University, Miami, FL 33199, USA; vmija002@fiu.edu (V.M.); jalci009@fiu.edu (J.A.)

**Keywords:** household, food waste, diet quality, grocery shopping, Hispanics, south Florida

## Abstract

The objective of this study was to explore the associations between food waste and the diet quality of foods purchased and with grocery purchasing behaviors. This was a cross-sectional study among 109 primary household food providers conducting primary shopping. Participants were recruited outside of local grocery stores and were asked to complete a survey assessing amounts of avoidable food waste and grocery purchasing behaviors. The diet quality of the foods purchased was assessed from grocery receipts using the Grocery Purchase Quality Index-2016 (GPQI-2016). Variables were associated using linear regression, analysis of covariance, and point biserial correlations. We found that fresh fruits (63%) and leafy greens (70%) were the foods that were the most wasted. The GPQI-2016 total score was significantly inversely associated with the total amount of food wasted (β  =  −0.63; 95% CI: −1.14,−0.12) after adjusting for important confounders. The reason “food past the date printed on the package” was directly correlated with food wasted (r = 0.40; *p* < 0.01) but inversely correlated with GPQI-2016 score (r = −0.21; *p* = 0.04). Food wasted, but not the GPQI-2016 score, was significantly higher among those who grocery shop 2–4 times per week compared to 1 time every 1–2 weeks (*p* = 0.02). In conclusion, food waste is inversely associated with diet quality and directly associated with grocery purchasing frequency.

## 1. Introduction

About a third of the worldwide food supply is wasted [1]. In the United States (U.S.), it has been estimated that 40% of the food produced ends up in landfills [2]. The production of food, which includes produce and livestock, requires resources such as energy, water, and land. It takes about 50% of U.S. land, 67% of its freshwater, and about 300 million barrels of oil per year to get food from the producer to the tables of consumers [2,3], yet only 60% of that food ends up being consumed [4]. A recent report shows that the annual cost U.S. consumer-level food waste was USD 240 billion [5]. Besides these costs, food waste represents a waste of key nutrients that are necessary to fill the nutritional gaps for millions of individuals [6]. It has been estimated that food waste accounts for about 1400 kcal per person per day in the U.S. [3]. Reducing food waste by only 30% and redirecting it to those who need it would be enough to feed the 42 million individuals in the U.S. who are food insecure [4]. Moreover, the food that ends up in landfills does not break down in the same way that it does in a compost pile, as it releases methane gas when it breaks down, contributing to greenhouse gas emissions [4,7]. Due to these negative consequences of food waste, there are several pledges around the world to reduce it. For example, European Parliament resolved to reduce food waste by 50% by the year 2030 [8]. The U.S. also set a goal to reduce food waste by 50% by the year 2030 [4].

An important part of reaching the goal of reducing food waste by 50% is implementing strategies at the household level to reduce food waste. It has been estimated that the average percentage of household food waste is 30–32% [5]. Therefore, research is needed at the household level to understand the predictors of food waste to better design and implement strategies for reducing it. There is already some light into the socio-demographic factors related to food waste at the household level. Those with a higher income may have a higher possibility of waste due to less concern with savings [9]. Some cultures value food re-utilization, which may result in less food waste [10]. Lack of education about food waste can also increase it [10]. Consumer behavior is another important factor. Studies in several countries have shown that buying more food than needed [11,12,13], planning shopping trips [12,13,14,15], and having a routine for using leftovers were important determinants of food waste [11,12,13,14,15,16]. However, all of these studies were done in Europe or Asia. In the U.S., an online survey of a nationally representative panel (*n* = 1010) found that the main reasons for wasting food were concerns about food safety (65%) and preferences for the freshest foods (60%) [17].

There is limited research relating food waste to diet quality [18,19]. These studies showed that higher diet quality is associated with greater food waste. This could be explained by the fact that fruits and vegetables, which are one of the main contributors to a high diet quality, make up about 1/3 of the food wasted [4,20,21]. Diet quality is often assessed by comparing the consumption of foods to dietary guidelines, using tools such as the Healthy Eating Index (HEI) [22,23]. The more compliant with the guidelines, the higher the diet quality score. Because some food groups are wasted in higher amounts than others, diet quality might be related to the amount of food that is wasted per household. Because of the research gap in the available data on food waste, diet quality, and shopping behaviors at the household level, this cross-sectional study was designed to evaluate the association between avoidable food waste and (1) the diet quality of the foods purchased and (2) consumer purchasing behavior among individuals in south Florida. It was hypothesized that a higher avoidable food waste was associated with the higher diet quality of foods that were purchased and that food waste was associated with grocery purchasing behaviors. The results from the present study may inform the development of specific strategies for reducing food waste.

## 2. Materials and Methods

### 2.1. Study Population

The cross-sectional study sample consisted of residents of Miami-Dade County in south Florida (in which 70% of the residents are Hispanic) who had just finished a primary shopping event. The inclusion criteria were being at least 18 years old, being the primary shopper of the household, having just completed a primary shopping event for the week (defined as purchasing most of the items needed by that household in this store), and allowing us to take a picture of the grocery receipt from that primary shopping event. There were no exclusion criteria other than not meeting the inclusion criteria.

Participants were recruited outside of local supermarkets such as Publix, Fresco y Mas, Sedano’s, and Aldi. After requesting permission from the supermarket manager to conduct the survey, the recruiters waited outside the store for potential participants. When a shopper had multiple items in their grocery cart, they were approached by the research team to ask if they would like to participate in a study about food waste. If they were interested, the research team explained the study, and they completed a short pre-screening questionnaire with the inclusion criteria. This was done either in English or Spanish, depending on the preferred language of the individual. If they qualified and agreed to participate, the research team (who was bilingual) provided a letter informing the participant about the study (informed consent was waived). The study was approved by the Institutional Review Board (IRB) of Florida International University (IRB-19-0158-AM02).

### 2.2. Survey

Participants completed a survey with the following sections:-Socio-demographics: Eligible participants were asked questions about their age, gender, household size, ethnic group, and an optional question about their weight and height, which was used to compute their body mass index (BMI);-Avoidable food waste: Participants completed a questionnaire that was based on the categories of food waste and reasons for food waste from the Household Food & Drink Waste questionnaire of the United Kingdom (U.K.) Waste & Resources Action Programme [24]. Based on the Food Loss and Waste (FLW) Protocol [25], handfuls were used to estimate frequency and quantity, as this is an easy unit that could be easily estimated by participants in a relatively short period of time. As a reference, participants were shown a picture of a hand holding an apple and a hand holding a handful of vegetables. The modified questionnaire included questions on categories of avoidable food waste and asked participants to estimate the frequency of food wasted in the last month as “times per month” and to estimate the usual amounts of the food wasted using “handfuls” as the unit of measure. Unavoidable food waste was described to participants as bones, banana peels, apple cores, onion peels, etc., and participants were asked only to report avoidable food waste. Therefore, this report is exclusively on “avoidable food waste”;-Consumer grocery purchasing behaviors: Grocery purchasing behaviors were also assessed using questions from the Household Food & Drink Waste questionnaire [24]. For grocery purchasing behavior, participants were asked to choose from the following options: “I buy almost all my food at a main shopping event”, “I buy some food at a main shopping event and then go back for smaller items”, and “I mostly buy food in smaller amounts and go often.” For grocery shopping frequency, participants were asked to choose from the following options: “3–4 times per week”, “2 times per week”, “1 time per week”, and “1 time every 2 weeks”.

### 2.3. Diet Quality of Foods Purchased

Diet quality was evaluated using the Grocery Purchase Quality Index-2016 (GPQI-2016), which is a validated method to assess the diet quality of foods purchased [26]. This index is modeled after the HEI-2010 [26,27]. It is based on the 29 food categories used in the United States Department of Agriculture (USDA) Food Plan [26]. These categories are re-grouped into the 11 components of the GPQI (Total Fruit, Whole Fruit, Total Vegetables, Greens and Beans, Whole Grains, Dairy, Total Protein Foods, and Seafood and Nuts, Refined Grains, Processed Meats, and Sweets and Sodas).

To calculate the GPQI-2016, each food and the amount paid for it was extracted from the grocery receipt and entered in an Excel spreadsheet using the 29 food categories. Using the Statistical Analysis System (SAS) code, shared by the developers of the GPQI-2016, the food categories were then re-grouped into the 11 components and a score was then assigned to each component based on the degree of adherence to the Dietary Guidelines as the ratio of observed to expected expenditure shares for each of these food groups. The 11 components are scored from a range of 0 to 5 or 10, for a maximum overall score of 75. The higher the score, the higher the overall diet quality.

### 2.4. Statistical Analysis

The sample size was calculated using OpenEpi V.3. For a power of 80%, an alpha level of 5%, and a hypothesized frequency of food waste of 21% [28], a sample of 109 individuals was needed. For descriptive statistics, frequency was used for categorical variables, and mean and standard deviation was used for continuous variables. To test the hypothesis that higher avoidable food waste was associated with the higher diet quality of the foods purchased (GPQI-2016 score), a linear regression was used, adjusting for age, gender, household size, ethnicity, and BMI. A linear regression was also used to associate the total food wasted with grocery purchasing behaviors and diet quality, in which the predictor variables were added to the model simultaneously. To test the other hypothesis that food waste is associated with reasons for food waste, a point-biserial correlation was used between the amount of food wasted with each reason for food waste. In addition, the amount of the total food wasted was compared to food purchasing behaviors using analysis of covariance, adjusting for age, gender, household size, ethnicity, and BMI. These analyses were also run using the diet quality (GPQI-2016 score). Data were analyzed using Statistical Package for the Social Sciences (SPSS) version 26. Significance was set at *p* < 0.05. Data are available upon request.

## 3. Results

### 3.1. Socio-Demographic Characteristics, Grocery Purchasing Behaviors, and Food Waste

A total of 109 participants were recruited outside of major supermarkets in Miami-Dade County in south Florida. The average age of the participants was 44.6 years old, with a mean of 3.2 individuals per household. Most were female (74%) and Hispanic (Cuban, Mexican, Venezuelan, Colombian, Dominican, Puerto Rican, and from other countries in Central and South America) (79%) (Table 1). Grocery purchasing was done mostly once (35%) or twice a week (29%) and most bought their food during a main shopping event (40%) or during a main event and going back for smaller items (44%).

Most participants self-reported wasting fresh fruits (63%) and fresh leafy greens (70%) (Table 2). On average, each of these foods was wasted 1.6 times per month, and the amounts wasted were 3.5 handfuls of fruits and 5 handfuls of leafy greens per month. Additionally, 38% of participants reported wasting grain products such as bread, rice, and pasta. However, this was the second most wasted, with participants throwing away an average of 3.8 handfuls a month. In total, the estimated amount of food waste was about 18 handfuls per month. The two most common reasons why people wasted food were because the food was spoiled (90%) or because it was past the date printed on the package (80%) (Table 3).

### 3.2. Association between Diet Quality and Food Waste

The association between the amount of food wasted and the overall diet quality of foods purchased is shown in Table 4. The GPQI-2016 total score was significantly inversely associated with the total amount of food wasted (β  =  −0.63; 95% CI: −1.14, −0.12), with food waste from other fresh vegetables (β  =  −0.13; 95% CI: −0.25, −0.01), grain products (β  =  −0.30; 95% CI: −0.55, −0.05), and dairy products (β  =  −0.09; 95% CI: −0.16, −0.01), after adjusting for age, gender, race/ethnicity, household size, and BMI. Results were also adjusted for total expenditure from the receipt of that purchase, and results were similar (data not shown).

### 3.3. Association between Reasons of Food Waste and the Total Amount of Food Wastes or Diet Quality

The associations between the reasons for food waste and the total amount of food wasted or the overall diet quality of foods purchased are shown in Table 5. Most reasons for food waste were directly significantly correlated with the total amount of food waste, with the highest correlation found for “food past the date printed on the package” (r = 0.40, *p* < 0.01). In relation to the diet quality of foods purchased, the reasons “cooked food too much but never served” (r = −0.22, *p* = 0.03) and “food past the date printed on the package” (r = −0.21, *p* = 0.04) were significantly inversely associated with the GPQI-2016 score.

### 3.4. Association between Grocery Purchasing Behaviors and Food Waste

Table 6 shows that the total amount of food wasted was significantly higher among those that grocery shop 2–4 times per week compared to 1 time every 1–2 weeks (*p* = 0.02). When the extreme groups were compared, those that grocery shop very frequently (3–4 times per week, *n* = 25 households) vs. those who grocery shop less frequently (2 times per month; *n* = 11 households), no significant differences were seen in total food waste (data not shown), although the sample size was too small in these groups to potentially detect differences. Additionally, no difference was observed in the total amount of total food waste by the method of purchasing (food purchased at a main shopping event, food purchased at a main shopping event and going back for smaller items, or food purchased only in small shopping events; *p* > 0.05). Additionally, no differences were observed for the GPQI-2016 total score by food purchasing behavior (*p* > 0.05). In addition, the GPQI-2016 total score (β = −0.241) and grocery purchasing frequency (β = 0.222) explained 12% of the variability of the total food wasted (F(2,88) = 5.868, *p* = 0.04, R^2^ = 0.12).

## 4. Discussion

Our results identified fresh leafy greens, fresh fruits, and grain products as being the most wasted food categories (in frequency and amount). Leafy greens made up about 28% of total food waste. When combined with the other fresh vegetables, total fresh vegetables made up about 40% of total food waste. This was followed by grain products and fresh fruits (20–21% of total food wasted). These results are in line with previous reports, in which fruits, vegetables, and grain products are the main avoidable food waste [2,19,21,29]. Additionally, these foods make up the largest percentages of total food waste in the U.S. [4,18].

We also found that the purchased foods with a higher diet quality was associated with lower total food waste and with the waste of other fresh vegetables, grain products, and dairy products but not with the amount of waste from fresh fruits and fresh leafy green vegetables. Because fruits and vegetables are one of the most important components of a high diet quality, it was hypothesized that as diet quality increased so would food waste. Although these food groups are known to promote health, few individuals in the U.S. consume enough of them [30]. However, the results of the present study did not align with this hypothesis or with the few reports available [18,19]. One of these reports is the study by Conrad and collaborators in 2018, which was a simulation “ecological” study using data from various U.S. government sources at the group level and not from data collected from households [18]. They found that higher diet quality was associated with greater food waste. However, because it was a simulation study, the results may not apply to associations at the individual level, and it may also be overestimating food waste. The other study was conducted by Carroll et al. among 85 Canadian families finding that the parents’ diet quality was directly associated with daily fruit and vegetable waste after adjusting for household income, although it was not significantly associated with total food waste [19]. They speculated that those with higher diet quality scores could be purchasing groceries more often, but this was not evaluated. Perhaps the associations found in the present study may be explained by food literacy, which has been defined as a collection of critical and functional knowledge, skills, and behaviors required to plan, manage, select, prepare, and eat food to meet needs and determine intake, which ultimately protects diet quality [31,32]. It could be argued that those with higher food literacy may be more aware of how to prepare and manage foods to prevent spoilage, which may lead to lower food waste. Future studies should evaluate if food literacy is related to food waste and why the waste of fresh fruits and fresh leafy greens was not related to the diet quality.

The present study also found a significant inverse association between food being thrown away because it was past the date printed on the package with the overall diet quality of the foods purchased but a direct association with the total amount of food waste. Because of time constraints, the differences in the meaning of the various dates printed on food products were not discussed with the participants. However, confusion about the various printed dates has been often reported in the U.S [2,4,33,34,35]. Throwing away food based on the date printed on the package and because of spoiled food may be related to individuals being concerned about the safety of the food [18,33]. More than 80% of consumers in the U.S. report throwing food away before the printed date due to confusion and concern for food safety [4]. Because fresh foods such as fruits, vegetables, and dairy products may have a shorter expiration date, throwing away these foods impacted the diet quality in this sample. Those with higher food literacy may have a higher knowledge on how to manage the expiration dates of the foods that they purchase. We also found a significant inverse association between cooked food never served with the overall diet quality of foods purchased and a direct association between cooked food never served, left on the plate, packaged food never opened, or spoiled food with the total amount of food wasted. Some participants reported verbally that they threw rice away every day because they did not like leftover rice. Other studies have reported that individuals prefer to eat freshly prepared foods and that they avoid consuming leftovers [11] and that the main reason for throwing away food is related to too much food being prepared and it not being possible to save the leftovers [36]. Other studies have also reported that the main reasons for wasting food were concerns about food safety (65%), preferences for the freshest foods (60%) [17], and because the food was spoiled [21,37]. Therefore, educational strategies may be needed in this group to learn how to use fresh foods before the expiration date, how to repurpose foods past their expiration date, how to cook enough food for the family, and how to store cooked foods properly. Because no other study has evaluated the associations between diet quality with reasons for throwing food away, more studies are needed in other populations to understand these associations.

A significant association between food waste and grocery purchasing frequency was also found in the present study. Those who purchased groceries 1–2 times per week or less had lower food waste compared to those that purchased groceries 2–4 times per week. Similar results have been found in a study conducted in Norway [38] and in Sweden [21]. However, an online survey conducted in Italy and Germany found a slight increase in food waste with decreasing shopping frequency in Germany, but in Italy, they found similar results to our study [37]. This may be related to better practices and routines when grocery shopping, as reported in a systematic review of household food waste practices in 2018 [39]. Careful planning of grocery purchasing (e.g., writing a shopping list, checking inventories, etc.) and purchasing frequency may reduce food waste, although not all the studies have shown this.

Our study had a few limitations. The fact that food waste was self-reported diminishes the reliability of the data. Ideally, food waste should be measured or weighed to accurately account for everything. In fact, there are differences in food waste amounts between studies using questionnaires to estimate food waste and studies using direct methods to quantify food waste. A study in Italy found that actual food wasted by weight measurements were higher by 1 kg per family per week compared to the estimates from self-reported questionnaires [40]. Studies only using direct methods found generally higher food waste (~3–4 kg per household per week [19,41,42]) compared to studies using only diaries (1.4–1.7 kg per household per week [21,43]). Therefore, future studies should confirm these results using direct methods. Although we attempted to survey across different supermarkets in Miami-Dade County, there was difficulty acquiring permission from many grocery managers to conduct the study, so a couple of grocery stores were frequented several times. Participation in farmer’s markets or Community Supported Agriculture (CSA) programs was not evaluated. The scarce data on CSA and food waste is contradictory, with a case study finding that overall food waste in CSA was lower than in supermarkets [44] but with another focus group finding that CSA participation led to food being wasted because too much was given [45]. This should be evaluated in future studies. Family income level was not assessed, which could have affected the associations as those with a higher income may have higher food waste [9]. Individuals with a lower income may purchase fewer fresh fruits and vegetables. Finally, those that chose to participate may have a higher awareness of the problem of food waste, and this could have affected our results. Nevertheless, the study has several strengths that are important to highlight. The study used the GPQI-2016, a validated scoring system for assessing the diet quality of the foods purchased in terms of compliance with the 2015–2020 Dietary Guidelines for Americans. Other diet quality measures rely on dietary recalls, which are biased by the participant’s lack of remembering all the of foods consumed in the previous 24 h. The grocery receipt may be considered a more objective way to evaluate diet at the household level. However, the GPQI-2016 score may be affected if the amount of food wasted in the household is large. This should be studied in future studies.

## 5. Conclusions

In conclusion, we found that a higher diet quality of foods purchased was associated with lower total food wasted and with the waste of other fresh vegetables, grain products, and dairy products. The diet quality score was inversely significantly associated with throwing food away because it was past the date printed on the package or because of too much cooked food that was never served. Additionally, food waste was directly and significantly associated with the following reasons: food past the date printed on food package, cooked food never served, left on the plate, packaged food never opened, or spoiled food. Finally, the total amount of food wasted was higher among those who grocery shop more frequently. These results may help tailor food waste reduction efforts for this group of primarily Hispanic adults in south Florida, which may be different from households in different countries. Educational interventions could be focused on the shelf-life of different foods, as food safety was one of the reasons why consumers sometimes throw food away before the date printed on food packages as well as on product use and the repurposing of ingredients to prevent food spoilage, particularly for fruits and vegetables.

## Figures and Tables

**Table 1 nutrients-13-02535-t001:** Socio-demographic characteristics and grocery purchasing behaviors of the sample (*n* = 109).

Characteristic	Mean ± S.D. or *n* (%)
Age (years)	44.6 ± 13.6
Household size	3.2 ± 1.2
Gender	
Female	76 (74%)
Male	27 (26%)
Ethnicity	
Hispanic	81 (79%)
Non-Hispanic White	15 (15%)
Non-Hispanic Black	0 (0%)
Non-Hispanic other	7 (7%)
BMI (kg/m^2^) *	27.6 ± 4.8
GPQI-2016 score	40.9 ± 9.64
Grocery purchasing frequency	
1 time every 2 weeks	12 (11%)
1 time per week	37 (35%)
2 times per week	29 (29%)
3 or 4 times per week	25 (25%)
Grocery purchasing type	
Most foods purchased at main shopping event	41 (40%)
Food purchased at main event + smaller events	45 (44%)
Food purchased only at small shopping events	17 (16%)

S.D. = standard deviation. * BMI was missing from 12 individuals.

**Table 2 nutrients-13-02535-t002:** Prevalence of food wasted and mean frequency and amount of food wasted per month by food category (*n* = 109 *).

Food Category	Prevalence of Food Waste %	Frequency (Times per Month) Mean ± S.D.	Amount of Food Wasted (Handfuls per Month) Mean ± S.D.
Fresh fruit (non-frozen or canned)	63	1.6 ± 2.2	3.5 ± 5.1
Starchy vegetables (yuca, potatoes, plantains, etc.)	19	0.4 ± 1.1	0.8 ± 2.3
Fresh leafy greens (spinach, romaine, herbs, etc.)	70	1.6 ± 1.9	5.0 ± 9.2
Other fresh vegetables (tomatoes, carrots, broccoli, etc.)	43	0.9 ± 1.5	2.1 ± 4.9
Grain products (breads, rice, pasta, etc.)	38	1.3 ± 3.7	3.8 ± 9.5
Meats and poultry	24	0.5 ± 1.1	0.7 ± 1.6
Dairy	32	0.4 ± 0.7	1.1 ± 2.8
Beans	10	0.2 ± 1.0	0.5 ± 2.0
Seafood	7	0.1 ± 0.5	0.2 ± 0.7
Total amount of food waste	−	−	17.7 ± 19.9

S.D. = standard deviation. * A total of6 participants did not complete all parts of the food waste section.

**Table 3 nutrients-13-02535-t003:** Number and percent of participants reporting specified reasons for food waste (*n* = 109 *).

Reason	N (%)
Spoiled food	93 (90%)
Food past the data printed on package	82 (80%)
Packaged food opened but not finished	56 (54%)
Cooked food never served	53 (51%)
Food left on plate after a meal	50 (49%)
Packaged food never opened	35 (34%)

* A total of 6 participants did not complete all parts of the food waste section.

**Table 4 nutrients-13-02535-t004:** Association between amount of food wasted and the diet quality of foods purchased (GPQI-2016 total score).

Food Group ^#^	Unadjusted β (95% CI)	Adjusted β (95% CI) *
Fresh fruits	−0.02 (−0.13, 0.09)	0.23 (−0.12, 0.16)
Fresh leafy green vegetables	−0.05 (−0.25, 0.16)	−0.08 (−0.34, 0.19)
Other fresh vegetables	−0.05 (−0.16, 0.06)	−0.13 (−0.25, −0.01)
Grain products (breads, rice, pasta, etc.)	−0.26 (−0.46, −0.06)	−0.30 (−0.55, −0.05)
Dairy	−0.05 (−0.11, 0.01)	−0.09 (−0.16, −0.01)
Total food wasted	−0.54 (−0.96, −0.12)	−0.63 (−1.14, −0.12)

* Adjusted for age, gender, household size, and body mass index (BMI). ^#^ Only the food groups that were wasted by at least 30 participants were included.

**Table 5 nutrients-13-02535-t005:** Point-biserial correlation between reasons for food waste and diet quality of foods purchased (GPQI-2016 total score) and total amount of food wasted (handfuls).

Reasons for Food Waste	Total Amount of Food Wasted (Handfuls)		GPQI-2016 Total Score	
	Pearson Correlation (r)	*p*-Value	Pearson Correlation (r)	*p*-Value
Cooked food never served	0.33	<0.01	−0.22	0.03
Food left on plate after a meal	0.38	<0.01	−0.18	0.07
Packaged food opened but not finished	0.19	0.06	−0.08	0.44
Packaged food never opened	0.25	0.02	−0.19	0.06
Food past the date printed on food package	0.40	<0.01	−0.21	0.04
Spoiled food	0.34	0.01	0.02	0.82

**Table 6 nutrients-13-02535-t006:** Comparison of total amount of food wasted or diet quality of foods purchased (GPQI-2016 total score) by grocery purchasing behaviors.

Variable	Amount of Food Wasted (Handfuls) Mean ± S.D.	*p*-Value *	GPQI-2016 Total Score Mean ± S.D.	*p*-Value *
Grocery purchasing frequency				
1 time every 1–2 weeks	12.8 ± 14.4	0.02	41.9 ± 9.65	0.383
2–4 times per week	22.5 ± 23.3		40.5 ± 8.10	
Grocery purchasing type				
Most foods purchased at main shopping event	17.1 ± 18.9	0.62	41.7 ± 9.27	0.740
Food purchased at main event + smaller events	15.6 ± 15.8		41.0 ± 8.37	
Food purchased only at small shopping events	23.3 ± 30.0		40.2 ± 8.55	

* Analysis of covariance adjusted for age, gender, household size, race/ethnicity, and BMI.

## Data Availability

Data are available upon request.

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
