# Peer review of "Food Waste and Its Association with Diet Quality of Foods Purchased in South Florida"

_nutrients, 2021, doi:10.3390/nu13082535_

Round 1
Reviewer 1 Report
This study examined the diet quality of foods purchased and the food waste behaviors of participants. Previous studies have examined consumers' diet quality and purchasing behaviors; however, this study's research question on food waste behavior is interesting. Food waste at the household level is a massive problem, and reducing food waste can mitigate environmental impacts and food insecurity.
Introduction: The introduction provides good background and significance of the study. Supporting studies on diet quality can be moved to the discussion section so that the introduction is short.
Discussion: Instructions for authors should be removed (line 242-245). It is more relevant to add Conrad et al. study to the discussion than in the introduction to understand better and compare the current study (261-262).
The authors have described very well the limitations of the study. It appears that the majority of the respondents are of Hispanic origin. It is unclear if the participants are fluent English speakers and whether the researchers administered the survey in English or another language.
Reviewer 2 Report
The paper provides some valuable information, and the diet quality from grocery receipt was a novel method. However, some texts and tables still need to be revised.
- Please define the abbreviations.
- Please follow the reference style suggested by the journal. For example, some ended up with “;”, and some online source lacked day accessed.
- As for the format of this article, please follow the “nutrient template” and put subsection titles as 2.1., 2.2. etc.
- Line 76: Please delete “higher” or use “higher parents’ diet quality” in line 75. Delete effect size (β =1.05; 95% CI:0.11, 1.99).
- “Line 87-89” is study aim. Please move/combine this paragraph to section “Introduction”. The study design “cross-sectional” could be moved to next paragraph.
- Line 112: Because the BMI was an optional question, how many missing data are they? (not in table 1)
- Line 114: In section of “avoidable food wastes”, the questionnaire was shortened from questionnaire of UK WRAP. However, the reference “24” was the final report of the household food and drink waste. I could not find any questionnaire in this report. Please clarify the origin. Besides, is there any objective measure to define “handfuls”? In general, I suggest defining more clearly about the reasons, frequency, category and amount of food waste in this section.
- Line 126: Please check the citation for footnote 17 and reformat the footnote itself for consistency.
- Line 135: For GPQI-2016, the original article showed that it was modeled after HEI-2010, not 2015. Is this a modified version? In addition, both articles (in reference 25 and 26) did not include alcoholic beverages and bottled water in GPQI. Please double check. Last, the main focus of this paper is GPQI-2016, not HEI-2015. In this section, I suggest to clarify the methods of grouping and scoring of GPQI. Too many information of HEI-2005 creates confusion. Besides, I don’t think their scoring approaches were similar (for example, as-consumer form vs. as-purchased form). You may move HEI to the section of discussion.
- Line 158: For power calculation, is there any reference to set food waste of 21%?
- Line 164-166: Grocery shopping behaviors did not refer to reason for food waste. Please confirm.
- In the section of “results”, I suggest adding subsection titles to make it clear.
- Line 177: the N(%) in the text was not consistent with them in the table. For example, 36% in the text but 35% in the table.
- Line 184-189: Please define the “proportion of sample wasting food” or describe it in the section of measures.
- Table 2: The sum of N=103 and 12 (participants who did not complete all parts of the food waste section) is not 109 (total participants of this study). Consider to put numbers in this table.
- Table 4: Should the title be “Association between the amount of food wasted and the diet quality of food purchased…”? Did you serve the amount of food wasted as independent variables and diet quality as dependent variables? Was the category of food wasted put in the model separately or together? Was there any correlation between food wasted categories that should be considered?
- Footer in table 4: The footer in table 4 did not show BMI in adjusted covariates. Please confirm. In addition, I suggest separating “adjusted for age, gender, and household size” and “Only the food groups that…” with different numbers of footers, and the numbers should be placed in the table accordingly.
- Line 205: Please explain the method to analyze for diet quality and grocery shopping frequency. I suggested moving line 204-206 to the last paragraph which explored the association between diet quality and grocery shopping frequency.
- Line 216:”food past the date printed on the package” (r=-0.40, p<0.01), but the correlation (r) was +0.4 in the table. Please confirm. The inconsistency was also noted in the abstract.
- Table 5. The words used in reasons for food waste should be consistent throughout the paper. For example, “Spoiled food” in table 3 but “food moldy or gone bad” in table 5.
- Line 230: I suggested discussing about the non-significant differences in the discussion.
- Table 6. The format of footnote should be consistent with other tables.
- Line 242-244. Please delete this paragraph.
- Line 255: What was the hypothesis based on? I suggested starting this paragraph from the finding of this paper, and then comparing it with previous studies and last, making discussion.
- Line 274-293: This paragraph focus on the reasons for wasting food but lack discussion about the association between reasons for wasting food and diet quality, which was the main finding of this paper. Some in-depth discussion may be required. The information in line 286-291 seemed repetitive. Please make it concise.
- Line 325 and line 327. The word “lastly” appeared twice.
- Line 327-334: It is about the sites of participation. I suggested moving this to the discussion in line 320-323, and making it concise and brief.
- As for the family income, maybe you could see if the total expenditure of that shopping affects the association between food wastes and diet quality.
- Please make the “conclusion” brief.
Round 2
Reviewer 2 Report
Review 2
While some errors were corrected after revision, they were quite avoidable and should have been addressed prior to submission. This applies especially to the accuracy of the methods and the main results. In general, more rigorous writing is required. Review and tighten the writing of your entire article to improve readability and understanding.
- Still not follow the template of this journal. For abstract: We strongly encourage authors to use the following style of structured abstracts, but without headings. For subsubsection, please use bulleted lists or numbered lists. Please review all sections.
- “P” in “P value” should be in italics.
- Most of the website links were lacking. Please review reference style.
- Line 63-77: The introduction was more concise after revision. However, you could still mention about research gap, such as lacking data of household level or shopping behaviors, or different study design in brief sentences to understand why this study was needed.
- Line 108: ref. 24 was not modified. When I searched WRAP website, “Questionnaires: Household Food & Drink Waste in the UK 2012” and “Final report: Household Food & Drink Waste: A product focus” were different files. Please confirm.
- 25 did not show the link.
- Line 108-112, Line 115-117: Please combine/simplify these sentences because they all talked about unit of “handful”. You could just keep “based on FLW [25]” but delete the introduction of FLW in the text to make it brief. I just wonder if all participants knew the meaning of “handful” to avoid bias. Was there any reference provided in the questionnaires, i.e., average handful is around the size of a baseball or other reference amount?
- Line 130: 2.2.2. not 2.2.3
- Line 141 and 143: Please make sure it was 31 or 29 categories.
- Line 130-146 2.2.3. GPQI: Please avoid repetitive information. “29 food categories” appeared for three times. Line 137-139 talked about scoring method. I suggest moving it to the second paragraph (line 140-146) where scoring method was introduced.
- Table 1. I suggest adding mean and SD of GPQI.
- I suggest using consistent wording to decrease confusion. For example, “grocery shopping habits” “food purchasing behaviors” “grocery purchasing behaviors” in Line 157, 160, 168 and “grocery shopping behaviors” in abstract. Please review them.
- Table 2. Please add “mean ± SD”. Besides, please define the abbreviation of “S.D.”.
- Table 4. I suggest moving the footnote “BMI was missing from 12 individuals” to table 1 and adding the information of BMI in table 1 to better understand the mean BMI of the participants.
- Line 235-236. I supposed that GPQI, grocery shopping frequency and total food wastes were put in the model simultaneously. The method described in line 156-157 was still not clear enough. Besides, the order of the description should better correspond to the order in the result.
- Line 237. R2àR2
- Suggest adding subtitle “Association between reasons of food waste and the total amount of food wastes or diet quality”.
- In general, the discussion is too long with repetitive information (i.e., line 253-256 and line 270-273, line 261-262 and line 264-265, line 280-281 and line 296-297, …) Food literacy is a good point. However, it appeared too many times in the discussion. Please make it concise and brief.
- Line 282-284: delete this sentence.
- Another point could also be considered: Could GPQI reflect actual diet quality if the amount of food wastes is large? GPQI is calculated from the foods purchased, in contrast, HEI was calculated from FFQ or dietary record or recall which reflects food actually taken.
- Line 305-307: Please make sure if ref [17, 21, 37] reported the main reasons for wasting food, or “the correlation or association between the reasons for food waste and amount of food wastes” which was the main focus of this paragraph. You may either delete it or move it to other paragraph that discuss about “reasons for food waste”.
- Line 328-337: These sentences all discussed about differences between studies using questionnaires and direct methods. There were too many detailed information. However, the focus was the “difference” between two methods. You could simplify which measurement was usually higher (about _____kg per week). Your result is underestimated or overestimated, and how you interpret it
- Line 364-365: Why were these three reasons chosen? (5 reasons correlated with total amount of food wasted in Table 5). Results were not necessarily all presented in the conclusion. Please consider deleting if it were not included in the main aims of your study.
Line 365-366: Why did the authors choose to present R2 in conclusion, rather than the main finding of table 6 (just like your conclusion in abstract). Besides, “lower frequency” and “lower score” were first presented here, but I could not find any beta coefficient in the results.
